# Recent Advances in the Treatment of Cerebellar Disorders

**DOI:** 10.3390/brainsci10010011

**Published:** 2019-12-23

**Authors:** Hiroshi Mitoma, Mario Manto, Jordi Gandini

**Affiliations:** 1Medical Education Promotion Center, Tokyo Medical University, Tokyo 160-0023, Japan; 2Service de Neurologie, Unité des Ataxies Cérébelleuses, CHU-Charleroi, 6000 Charleroi, Belgium; mmanto@ulb.ac.be (M.M.); jordig85@gmail.com (J.G.); 3Service des Neurosciences, University of Mons, 7000 Mons, Belgium

**Keywords:** cerebellar ataxias, therapies, motor rehabilitation, cognitive rehabilitation, non-invasive cerebellar stimulation, neurotransplantation

## Abstract

Various etiopathologies affect the cerebellum, resulting in the development of cerebellar ataxias (CAs), a heterogeneous group of disorders characterized clinically by movement incoordination, affective dysregulation, and cognitive dysmetria. Recent progress in clinical and basic research has opened the door of the ‘‘era of therapy” of CAs. The therapeutic rationale of cerebellar diseases takes into account the capacity of the cerebellum to compensate for pathology and restoration, which is collectively termed cerebellar reserve. In general, treatments of CAs are classified into two categories: cause-cure treatments, aimed at arresting disease progression, and neuromodulation therapies, aimed at potentiating cerebellar reserve. Both forms of therapies should be introduced as soon as possible, at a time where cerebellar reserve is still preserved. Clinical studies have established evidence-based cause-cure treatments for metabolic and immune-mediated CAs. Elaborate protocols of rehabilitation and non-invasive cerebellar stimulation facilitate cerebellar reserve, leading to recovery in the case of controllable pathologies (metabolic and immune-mediated CAs) and delay of disease progression in the case of uncontrollable pathologies (degenerative CAs). Furthermore, recent advances in molecular biology have encouraged the development of new forms of therapies: the molecular targeting therapy, which manipulates impaired RNA or proteins, and the neurotransplantation therapy, which delays cell degeneration and facilitates compensatory functions. The present review focuses on the therapeutic rationales of these recently developed therapeutic modalities, highlighting the underlying pathogenesis.

## 1. Introduction

### 1.1. History

Cerebellar ataxias (CAs) encompass a heterogeneous group of disorders characterized by motor incoordination, affective impairment, and disorganized cognitive operations resulting in dysmetria of thought [1,2]. CAs are often disabling, impacting significantly on the daily life activities of affected patients, from the beginning of life to the elderly. With the aging of the population and the discovery that cerebellum participates in numerous motor functions and cognitive operations at all stages of life, CAs are growingly recognized worldwide. Since the seminal contribution of Gordon Holmes [3] on motor deficits one hundred years ago, many new cerebellar disorders have been identified. Furthermore, physiological studies have clarified the mechanisms underlying CAs and involving neural circuitries since 1960s. In particular, molecular biology methods have elucidated the genetic deficits (more than 140 cerebellar disorders are associated with gene mutations) and molecular changes underlying cell death since 1980s.

During the last three decades, considerable advances have been made in the field of CAs therapy [4]. For example, evidence-based therapeutic strategies have been established for metabolic and immune-mediated CAs. Improved protocols have been proposed also in the field of rehabilitation to reconstruct or compensate lost cerebellar functions. On the other hand, sufficient evidence indicates that non-invasive cerebellar stimulation modulates excitability in the residual cerebellar circuits to improve symptomatically CAs. Furthermore, rapid developments in novel technologies currently offer a real possibility of stopping neuronal degeneration. Typical examples are molecular targeting therapies designed to manipulate RNAs or key proteins and the transplantation of grafted stem cells to facilitate the reversibility of host cells. Thus, these clinical and basic advances have opened the door to a new era where neuroscientists and clinicians will control the process of cell death and restore impaired cerebellar functions [5,6]. The aim of the present review is to assess the newly established therapeutic methods and anti-ataxic drugs.

### 1.2. Therapeutic Strategies Based on Cerebellar Reserve

The cerebellum is endowed with the capacity to compensate and restore damage inflicted by various pathologies, a specific feature in the Central Nervous System (CNS) [4,7]. We defined this characteristic ability as cerebellar reserve [4,7]. Cerebellar reserve can be physiologically attributed to the ability to update the internal models, the neural mechanisms that mimic the input/output characteristics of the motor apparatus [8,9]. The same concept applies also for social defects, affective dysregulation, and cognitive operations under cerebellar control. This is a unique feature in the brain. The forward internal model, which can predict sensory consequences in response to issued motor commands, is embedded in the cerebellum [10]. The cerebellum receives a copy of the motor commands (corollary discharge) (Figure 1A). The cerebellar circuitry is heavily connected with thalamic nuclei and cerebral cortex, in particular thanks to multiple cerebello-cerebral loops running in parallel. The internal model, once acquired through motor learning, can be updated facing an injury, so that novel predictions of sequences are implemented [11,12]. Physiologically, two mechanisms operate. First, many forms of synaptic plasticity are active in the cerebellar cortex [4,7]. Second, mossy fibers project to diverse microzones, the functional units in the cerebellar cortex, implying that one microzone receives redundant information conveyed by mossy fibers from the cerebral cortex and the periphery [4,7]. In other words, the cerebellar circuit has unique anatomical and functional properties allowing a reorganization of its modules.

During a period when cerebellar damages remain to be functional in synaptic transmissions or ion channels, cerebellar reserve is preserved. As cell death advances, cerebellar reserve is lost (Figure 1B,C is a schematic diagram of the cerebellar reserve) [6,7]. As disease progresses, cerebellar reserve is lost, impacting on motor/cognitive/affective control. The clinical courses of many cerebellar diseases suggest the existence of a therapeutic threshold [4]. Above this threshold, treatments may halt the progression of the disease, leading to the resolution of symptoms. Later, although treatment can effectively stop the progression of the disease, the CAs remain phenotypically unchanged. Thus, early therapeutic interventions should be applied during the period when cerebellar reserve is still preserved. In other words, clinicians should not miss the window of opportunity that offers specific advantages to the cerebellum. Therapeutic interventions can be classified into two modalities: cause-cure treatments and neuromodulation therapies [4,5]. The former aims at stopping the progression of the disease, whereas the latter aims at repairing lost cerebellar functions. The clinical output of each modality will depend on the nature of the underlying pathology. When the pathology is potentially controllable (e.g., metabolic and immune-mediated CAs), combinations of cause-cure treatment and neuromodulation therapies will result in recovery from CAs. On the other hand, when the pathology is progressive and uncontrollable (e.g., degenerative CAs), the combination therapies will prolong the process of disease progression. In summary, the notion of cerebellar reserve provides a framework for therapeutic strategies.

## 2. Cause-Cure Treatment; Treatments Designed to Prevent Disease Progression

### 2.1. Metabolic Cerebellar Ataxias

Therapies for metabolic CAs can be classified into three categories; abstinence (e.g., alcohol), replacement (e.g., vitamin B1), and use of chelates (e.g., iron accumulation).

#### 2.1.1. Alcohol-Related Cerebellar Ataxias

Chronic alcoholism is associated with cerebellar atrophy, especially the anterior superior vermis, the severity of which correlates with that of gait ataxia [13,14]. Two pathomechanisms have been considered, the associated global malnutrition and direct toxic effect of ethanol [14]. Abstinence and improvement of nutrition are the basic strategy of treatment, complemented by rehabilitation.

#### 2.1.2. Vitamin B1 Deficiency (Wernicke’s Encephalopathy)

Wernicke’s encephalopathy is caused by thiamine (vitamin B1) deficiency and is characterized by the triad of sudden-onset changes in mental status, cerebellar symptoms, and ophthalmoplegia. Wernicke encephalopathy is sometimes misdiagnosed, leading to persistent dysfunctions and, in some patients, death. [15,16]. The irreversible lesions in the thalamus and mammillary bodies lead to the development of Korsakoff syndrome, which is characterized by persistent anterograde and retrograde memory loss, confabulation, and disorientation [17]. Korsakoff syndrome is observed in alcoholic patients more frequently than nonalcoholic patients [15]. The prognosis depends on early administration of thiamine [18]. Thus, Wernicke’s encephalopathy requires urgent medical attention, and thiamine should be administrated in patients suspected with the disorder, either intravenously or intramuscularly, rather than orally [15,16]. In this regard, there is still no consensus on the optimal dose of thiamine [16]. While a dose of 100 mg/day thiamine has been recommended [19], various other regimes have also been proposed (Table 1) [15,19,20]. Nevertheless, such treatment should be continued until no further clinical improvements are observed [16].

#### 2.1.3. Iron Deposition: Superficial Siderosis

Superficial siderosis is characterized pathologically by deposition of hemosiderin, an iron-containing compound, in the superficial layers of the CNS in close proximity to the cerebrospinal fluid (CSF), including the subpial and subepidermal area [21,22]. Patients exhibit a slowly progressive CA, sensorineural hearing loss and/or symptoms of myelopathy. The causal therapeutic strategy is surgical ablation of the bleeding source. However, the therapeutic benefits during the long-term course are uncertain despite clearance of the CSF [21]. Another strategy is iron chelation. Desferrioxamine has been used in a few patients without any effects [23]. Another iron chelator, deferiprone, has been the focus of interest due to its high transparency to the blood-brain barrier (BBB) [24,25]. Haematological monitoring is required due to the risk of neutropenia with sepsis.

### 2.2. Immune-Mediated CAs

Immune-mediated CAs include many subtypes, which include gluten ataxia (GA), paraneoplastic cerebellar degeneration (PCD), post-infectious cerebellitis (PIC), and anti-GAD ataxia [26,27,28,29]. Despite the diversity, treatment is based on common strategies; (1) when autoimmunity is triggered by another condition (e.g., gluten sensitivity for GA, neoplasm for PCD, and infection for PIC), the first priority goes to removal of the underlying trigger, (2) in case of no benefits or when autoimmunity is not activated by another condition, immediate immunotherapy is recommended [28,29]. Immunotherapies include corticosteroids (intravenous methylprednisolone for induction therapy and oral prednisolone for maintenance therapy), intravenous immunoglobulins (IVIg), immunosuppressants (e.g., mycophenolate mofetil, cyclosporin, and cyclophosphamide), plasmapheresis, and rituximab, either in alone or in combinations [26,27,28,29]. There are no large-scale randomized clinical trials on immunotherapies in these types of CAs. Standard therapies are summarized in Table 1.

#### 2.2.1. Gluten Ataxia (GA)

GA is characterized by gluten sensitivity with or without enteropathy [30]. Gluten sensitivity is assessed by measuring the presence of anti-gliadin, transglutaminase 2, and 6 Abs [26,27,28,30]. The golden standard treatment is gluten-free diet (GFD) which eliminates the antigen causing gluten sensitivity [31]. One large-scale study by Hadjivassiliou et al. [31] showed improvements of CAs in patients who adhered to strict GFD, with associated reduction in titer of anti-gliadin Ab, whereas those who did not receive the GFD therapy showed worsening of their condition. It is now considered that the lack of therapeutic benefits by GFD is probably due to poor adherence to the diet or hypersensitivity to gluten, where a small amount of gluten still present in the usual gluten-free food causes strong immune-mediated response [31]. Thus, measurement of anti-gliadin Ab level can re-enforce strict adherence to GFD or adherence to wheat-free diet, before switching to immunotherapy, such as IVIg or immunosuppressants (e.g., mycophenolate mofetil) [32,33].

#### 2.2.2. Paraneoplastic Cerebellar Degeneration (PCD)

PCD is characterized by autoimmunity towards the neoplasm [34,35,36]. The diagnosis of PCD is based on (1) rapid diagnosis of cancer, which usually develops within five years of diagnosis of CA, or (2) the presence of well-characterized onconeuronal antibodies (e.g., anti-Yo, anti-Hu, anti-CV2, anti-Tr, anti-Ri, anti-MA2) [34,35,36]. Immediate treatment of the neoplasm should be the first objective of treatment in order to prevent systemic metastasis and remove antigens that trigger the autoimmunity, followed by immediate induction of immunotherapy using corticosteroids (intravenous methylprednisolone or oral prednisolone), IVIg, plasmapheresis, immunosuppressants, and/or rituximab, either alone or in various combinations [35,37,38,39]. However, there are no reports of significant differences in the response to these types of immunotherapies. The anti-neoplasm therapies and immunotherapies have no benefits in most PCD cases, with a bleak prognosis (relatively short median survival time: 10.2 to 43 months) [40,41]. Nevertheless, some PCDs are stopped when the neoplasm is quickly treated. Patients enter in remission and may exhibit a mild to severe residual cerebellar syndrome, for instance in anti-Tr PCD. Retrospective studies involving a few responders have identified certain prognostic factors (e.g., anti-Tr or anti-Ro antibody) [42].

#### 2.2.3. Post-Infectious Cerebellitis (PIC)

PIC affects mainly children and is triggered by infection, usually a viral infection, most commonly varicella [43,44]. PIC is usually a self-limiting disease. One large study of 60 pediatric patients who were followed for more than four months, showed that 72% of the patients showed complete recovery form PIC [45]. Thus, conservative treatment and close observation are recommended, although intravenous acyclovir reduces the duration of the infection if administered very early. Immunotherapy should be used only in cases with persistent symptoms (with corticosteroids, IVIg, immunosuppressants, and/or plasma exchange, either alone or in combination) [4]. Antibiotics are administered for PIC associated with Lyme disease, especially ceftriaxone. Minocycline is used for cerebellitis associated with Coxiella brunetti infection Transtentorial or transforaminal herniations are rare complications requiring posterior fossa decompression and external ventricular drainage.

#### 2.2.4. Anti-GAD Ataxia

This type of CA is associated with high titers of anti-GAD65Ab in both serum (10–1000 fold, compared to patients with type 1 diabetes mellitus) and CSF [46]. Anti-GAD Ab is assumed to play a pathogenic role, and is sometimes associated with epilepsy or stiff-person syndrome [47,48]. The exact conditions that trigger autoimmunity against GAD65 are unknown. Accordingly, treatment is geared towards: (1) induction to minimize CAs and (2) subsequent maintenance therapy to prevent disease progression [4,5]. Both modalities include various immunotherapies, ranging from corticosteroids, IVIg, immunosuppressants, plasmapheresis, and rituximab, either alone or in combinations [4,5]. No significant difference has been reported among these options. The prognosis is better in the subacute type than the chronic type [49].

### 2.3. Autosomal Recessive Cerebellar Ataxias (ARCAs)

This heterogeneous group of conditions gathers disorders generally manifesting before the age of 30 with progressive gait ataxia [50,51]. ARCAs often exhibit extra-cerebellar neurological symptoms or even extra-neurological deficits. The diagnosis of ARCAs may turn to be very challenging at the beginning of the disease [50,51]. Newly developed therapies designed to manipulate cellular functions have so far failed to show significant improvement in ARCAs. However, chelation or replacement therapy is sometimes beneficial especially where metabolic changes lead to accumulation or deficiency of a particular substance (Table 2).

#### 2.3.1. Friedreich’s Ataxia (FRDA)

Symptoms of FRDA are related to spinocerebellar degeneration, cerebellar pathology, and peripheral nerve damage [50,51]. Guanine-adenine-adenine (GAA) trinucleotide repeat expansion is inserted in the *frataxin* (*FXN*) gene, located on chromosome 9q13 [57]. Frataxin is a mitochondrial protein implicated in iron metabolism, oxidative stress, energy metabolism, and other mitochondrial functions [50]. The initial treatment strategy was to increase mitochondrial function using a combination of antioxidants (e.g., vitamin E, coenzyme Q10 [58], and idebenone [52,53]). However, the treatment failed to demonstrate substantial improvement. The second approach was removal of accumulated iron. Treatment with deferiprone, an iron chelator [54], showed deterioration of the condition and proved inconclusive. The third choice was interferon-γ, a cytokine implicated in iron metabolism, in an attempt to increase the expression level of frataxin protein [55]. One open-label study showed that treatment with interferon-γ resulted in improvement in FRDA score [55].

#### 2.3.2. Ataxia-Telangiectasia (AT)

A-T is a multi-organ disorder caused by mutations in the ataxia–telangiectasia mutated gene (*ATM*) on chromosome 11q22.3 [56]. Patients develop CAs and immune deficiency during childhood [50,51]. The *ATM* protein is implicated in the coordination of cellular response to DNA double-strain breaks and in oxidative stress [50]. One case report highlighted the potential benefits of betamethasone [59]. The results were subsequently confirmed in a one-month randomized clinical controlled trial [60]. Although betamethasone is known to have anti-oxidant properties, the long-term safety remains to be tested [50].

#### 2.3.3. Ataxia with Vitamin E Deficiency (AVED)

The clinical manifestations of AVED resemble those of FRDA [50,51]. AVED is caused by mutations in the α-tocopherol transfer protein gene (*TTPA*) on chromosome 8q13.1 [61]. Since this protein is involved in the production of low-density lipoproteins in the liver through incorporation with α-tocopherol (vitamin E), the systemic supply of vitamin E is impaired in AVED. Replacement of vitamin E is an effective treatment for AVED [50].

#### 2.3.4. Abetalipoproteinemia

Abetalipoproteinemia is caused by mutations in both alleles of the gene for microsomal triglyceride transfer protein [51]. The disorder causes hypocholesterolemia, malabsorption of fat and fat-soluble vitamins (vitamin A, E, D, K) leading to coagulopathy, retinal degeneration, neuropathy, and ataxia [51]. Treatment is based on low fat diet, decreased long-chain fatty acids and oral essential fatty acids. Vitamins are supplemented orally (vitamin E 100–300 IU/kg/day, vitamin A 100–400 IU/kg/day, vitamin D 800–1200 IU/kg/day, vitamin K 5–35 mg/week).

#### 2.3.5. Wilson’s Disease (WD)

WD is caused by mutations in *ATP7B* gene which encodes a copper-transporting P-type ATPase [50,51]. Untreated WD will occasion liver cirrhosis accompanying with a severe neurologic disorder [50]. This disorder of copper metabolism is treated with D-penicillamine (1–2 g/day), trientine (15–20 mg/kg daily), and zinc acetate/sulfate (50–250 mg/day). Liver transplant is considered in the fulminant form.

#### 2.3.6. GLUT1 Deficiency

GLUT1 deficiency is caused by a mutation in *SCL2A1* gene [51]. Cerebellar ataxia is part of a complex phenotype including seizures, developmental delay, microcephaly and spasticity. Glucose levels are decreased in the CSF. Treatment is based on ketogenic diet.

#### 2.3.7. Refsum’s Disease (RD)

RD is caused by mutations in either phytanoyl-CoA hydroxylase (*PHYH*) or PTS2 Receptor (*PEX7*) gene, which results in accumulation of phytanic acid (PA) [62,63]. The classical form of RD is characterized by retinitis pigmentosa, anosmia, polyneuropathy, deafness, ataxia, cardiac arrhythmias, and ichthyosis. The established therapy is dietary restriction of PA-containing foods such as animal fats and green vegetables [50,51]. Patients should avoid rapid weight loss and fasting, which activate lipolysis and accumulation of PA [50]. Plasmapheresis is also recommended for acute presentation [50].

#### 2.3.8. Cerebrotendinous Xantomatosis (CTX)

CTX is a disorder of bile metabolism caused by mutations in *CYP27A1* gene on chromosome 2q33 [64]. The initial clinical features include cerebellar deficits, parkinsonism, dystonia, upper motor neuron weakness, epilepsy, intellectual disability and dementia, psychiatric symptoms, and peripheral neuropathy [50,51]. Extra-neurological deficits include diarrhea, cataract, xanthomas and premature atherosclerosis. The gene encodes a mitochondrial sterol 27-hydroxylase, which is involved in the formation of bile acid. The impairment in 27-hydroxylase interferes with the formation of bile acid, leading to accumulation of cholesterol and cholestanol, the latter of which shows neural toxic actions. Replacement of the decreased bile acid elicits negative feedback on activated status in the bile formation pathway, resulting in decreased flow toward cholesterol. Based on these abnormalities, chenodeoxycholic acid, ursodeoxycholic acid, cholic acid, and taurocholic acid have been used with a positive response [50].

#### 2.3.9. Niemann-Pick Disease Type C (NPC)

NPC is caused by mutations in *NPC1* or *NPC2* genes, which encode intracellular cholesterol transporters [65]. The juvenile form is typically characterized by CA in association with movement disorders, dysphagia, vertical supranuclear ophthalmoplegia, and cataplexy [50,51]. The impairments in *NPC1* or *NPC2* genes lead to accumulation of cholesterol and glycosphingolipids [50,51]. Miglustat, an inhibitor of glucosylceramide synthesis, is the only approved medication with recognized efficacy in relieving neurological symptoms [66].

### 2.4. Episodic Ataxias (EAs)

EAs are characterized by recurrent attacks of vertigo and CA lasting up to a few hours [67]. Attacks are attributed to mutations in the *CANCNA1A* gene encoding the α-subunit of a P/Q-type calcium channel [68]. EA type 2 is the most frequent form [67]. Most patients show oculomotor disturbances including gaze-holding deficits, smoot pursuit, down beat nystagmus (DBN), even outside of the attack. A case series of four patients showed that 4-aminopyridines (4-AP), a nonselective blocker of the Kv family of K channels, decreased the number of attacks [69]. Subsequently, a randomized control study confirmed not only a reduction in attack number but also a decrease in attack time and improvement of severity of CA [70]. 4-AP is mainly a blocker of the Kv1.5 voltage-activated potassium channels. Thus, it prolongs the duration of action potentials in axons because of delayed repolarization, which could induce larger Ca^2+^ influx, compensating the reduced P/Q-type Ca^2+^ current density associated with EA2 mutation [71]. 4-AP is also effective against down-beat nystagmus (DBN) encountered in various pathologies [72,73].

## 3. Neuromodulation Therapies: Therapies Enhancing the Cerebellar Reserve

### 3.1. Motor Rehabilitation

Ample evidence suggests that conventional motor rehabilitation is effective in patients with limited lesions (e.g., cerebral infarction, hemorrhage, or trauma). However, this is uncertain in patients with degenerative CAs of progressive nature. Two recent large-scale and case-control designed studies demonstrated the therapeutic benefits of motor rehabilitation [74,75,76] (Table 3). In these studies, intensive four-week motor rehabilitation improved items of stability and limb coordination in Scale for the Assessment and Rating of Ataxia (SARA) score [74,76] and in the score related to activities of daily living (ADL), as determined by the functional independence measure (FIM) [76]. These protocols included intensive whole-body coordinative training on balance and mobility function. At 12 months after the cessation of intensive motor rehabilitation, assessment of the participating patients who performed exercises continuously during the study period, demonstrated the beneficial effects of the rehabilitation therapy during the natural progression of the diseases [74].

### 3.2. Cognitive Rehabilitation

In 1998, Schmahmann and Sherman described the cerebellar cognitive affective syndrome (CCAS or Schmahmann syndrome), which is characterized by executive dysfunctions, impaired visuo-spatial cognition, personality changes, and language deficits [77]. Overall, patients exhibit a dysmetria of thought and impaired affect. There are no systemic studies on cognitive rehabilitation [78]. Since there are only a few case reports that confirm the notion of Schmahmann syndrome [78,79,80,81,82], appropriate rehabilitation protocols and their outcomes remain unclear [78] (Table 3).

The clinical features of Schmahmann syndrome are subtle in adults and resemble those of cortical cerebral lesions. However, strictly speaking, it should be acknowledged that the rehabilitation programs designed for patients with cerebral disorders cannot be applied in patients with Schmahmann syndrome [78]. Schmahmann hypothesized that “unlike cortical deficits, cerebellar deficits might be compensated for, in part, by bringing the issue at hand to conscious awareness, focusing on the problem in order to address it” [83]. Consistently, the method of conscious behavior, instead of automatic behavior, was reported to be effective in some patients [80].

### 3.3. Non-Invasive Cerebellar Stimulation

Recent studies have concluded that non-invasive cerebellar stimulation (e.g., transcranial magnetic stimulation (TMS) and transcranial direct current stimulation (tDCS)) improves motor deficits in CAs [84,85] (Table 4). Studies with an active/sham-controlled design (six/seven studies) showed that non-invasive cerebellar stimulation improves the clinical scores of SARA and ICARS [86,87,88], gait ataxia measured using 8–10 m walking test [89,90,91], limb ataxia assessed using nine-hole peg test [86,87,88], tremor [86,87,88], blood flow in the cerebellar hemisphere [89], and electroencephalogram [92] in ataxic patients. The improvement was not limited to a short period (immediately after the last session), but was rather a durable effect after administration of several sessions (4–12 weeks after the last session). Notably, improvements in gait ataxia and dysmetria are associated with amelioration of exaggerated long-latency stretch reflexes [93] and impaired predictive activities of antagonistic muscles in fast goal-directed movements [86,92], respectively. These results show improvement in elemental impairments associated with CAs. On the other hand, the extent of the above improvements remains moderate (e.g., 3%–10% in SARA and 12% in ICARS) [87,88], which highlights the need for the development of more efficient protocols.

Functional reorganization of the cerebellar motor control appears to underlie the abovementioned therapeutic benefits of non-invasive cerebellar stimulation. The following findings could provide clues to the possible mechanisms of action:A single magnetic pulse inhibits the amplitude of motor evoked potentials, an index of excitability of the primary motor cortex, which is termed cerebellar brain inhibition (CBI) [94]. Thus, activation of Purkinje cells (PCs) inhibits the excitatory facilitation of the dentato-thalamo-cortical pathways [84,85]. On the other hand, tDCS modulates spontaneous neural activities at the target site through constant electrical current during a particular period, generally 20 min [84,85,95]. This sustained modulation of excitability in the cerebellar cortex changes CBI in a polarity-dependent manner [84,85]; i.e., anodal tDCS increases CBI, whereas cathodal tDCS decreases CBI [96,97]. It should be acknowledged that the long-term therapeutic effect corroborates plasticity changes in the cerebellar cortex.A previous physiological study [98] showed that activation of neurons of the dentate nucleus induced by diminished inhibition from PCs (i.e., disinhibition) facilitates the execution of a particular movement, while suppression of the dentate nucleus neurons by increased PC activity (i.e., inhibition) contributes to the stabilization of unnecessary movement. Thus, the cerebellum serves as a predictive modulator through disinhibition/inhibition of the dentato-thalamo-cortical pathway. Thus, damage of the cerebellar circuits could impair the formation of disinhibition/inhibition, thus leading to asthenia (impairment of initiation) and adventitious movements (impairment of stabilization), respectively [99].The tDCS-induced improvement was associated with facilitation of CBI [88].

Taken together, the therapeutic rationale for non-invasive cerebellar stimulation seems to be based on long-lasting restoration of inhibitory modulations by PCs on the dentato-thalamo-cortical pathways. Repair of PC activity is necessary not only for inhibition but also for disinhibition, since suppression of tonic PC activities is a prerequisite for the generation of disinhibition. Thus, non-invasive cerebellar stimulation that can enhance CBI might appropriately modulate activities of the cerebellar output, so as to repair CAs.

There is a consensus that targeting the cerebellum might become an effective approach to modulate cognitive activities of remote cerebral cortex [78]. For example, right cerebellar tDCS improved language recovery in infarction-induced aphasia and anarthria [100]. However, it is uncertain whether non-invasive cerebellar stimulation improves the genuine cerebellar-induced cognitive symptoms (CCAS) [78,101,102].

## 4. Novel Therapies

### 4.1. Recent Advances in Treatments of Autosomal Dominant Cerebellar Ataxias (ADCAs)

Autosomal dominant CAs (ADCAs) are classified into two groups on the basis of their genetic mutations, ADCAs caused by microsatellite repeat expansions and ADCAs caused by point mutations [103]. The former is furthermore categorized into two types, ADCAs induced by polyglutamine-coding CAG repeat expansions and those induced by non-protein-coding repeats [103]. Recent advances in molecular manipulation methods have enabled the development of disease-modifying drugs for ADCAs.

#### 4.1.1. Polyglutaminopathies

The cause of neural degeneration in spinocerebellar ataxias (SCA) type 1, 2, 3/Machado-Joseph disease, 6, 7, 17 and DRPLA [104] is DNA mutation of an expanded glutamine-encoded CAG (cystine-adenine-guanine)-repeat sequence. The pathology includes expansion of the CAG-repeat, leading to abnormal proteins that contain long polyglutamine (polyQ) tract, and subsequent aggregation of the misfolded polyQ, forming specific cytoplasmic and nuclear inclusions [104]. So far, the outcome of treatment with compounds designed to rectify the impaired molecular or cellular functions has not been successful. However, the following molecular interventions are promising future therapies.

*Clearance of misfolded proteins*: Misfolded proteins caused by polyglutamine expansion are degraded by the ubiquitin-proteasome system (UPS) [104]. It is widely accepted that aggregation of polyQ-containing proteins impairs UPS [105], rendering the latter unable to clear abnormal accumulation of a variety of toxic proteins, which results in cellular dysfunction and/or apoptosis [104]. Under these conditions, activation of chaperone, which induces natural folding equilibrium, is a potentially promising therapeutic target. Consistently, it was shown that certain chaperones, such as members of the Hsp70 family, attenuated aggregation of polyQ and the associated cell toxicity in experiments using fly or mouse models [106]. One alternative candidate is activation of the autophagy-lysosome system, which is involved in routine recycling of cytoplasmic components and known to protect neurons from accumulation of misfolded proteins [104]. It has been reported that administration of beclin 1, an autophagy effecter protein, improved motor incoordination in SCA type 3 model mouse [107].

Nascent polypeptide associated complex (NAC), a ribosome-associated protein biogenesis factor (particularly the N-terminal peptide) inhibits the aggregation of polyQ expanded proteins, associated with several types of SCAs [108]; therefore this approach represents also a promising therapy for SCAs.

*Re-regulation of transcription*: Expansion of polyQ impairs the transcription mechanisms through interaction with transcriptional proteins and DNA or with chromatin remodeling [104]. Histone acetyltransferases (HATs) is one of affected regulatory proteins [109]. HATs regulate the transcriptional competence of chromosomes, temporal promotor activity, and protein activation/inactivation [104]. It was shown that valproic acid, a histone deacetylases inhibitor, assumed to restore acetylation/deacetylation balance, rescued polyQ-mediated toxicity in SCA type 3 model cells and mouse [110].

*Therapies targeting DNA and RNA**—Antisense oligonucleotides (ASOs):* Antisense oligonucleotides (ASOs) have an RNA-like structure. In gain-of-function disorders, the main purpose is to decrease the mutated mRNA expression by use of single-stranded ASOs, by blocking translation or by a direct silencing effect [111]. In particular ASOs bind complementary mRNA using Watson-Crick hybridization; this process leads to RNase H enzymes recruitment [112]. In SCA3 fibroblasts the removal of central 88 amino-acid region of the ataxin-3 protein was realized with ASOs [112]. In a mouse model of SCA2 the administration of intrathecal ASOs decreased the levels of the mutated protein, improving the firing of Purkinje neurons as well as motor tasks [113]. Similarly mice receiving ASOs show a slower progression of motor deficits and longer survival [114]. Comparable findings have been described in the SCA3 mouse model [115]. In 2017 Moore et al. and Toonen et al. [116], using the same model, have demonstrated a reduction of 50% of mutated ataxin-3 in the cerebellum, diencephalon, forebrain and cervical spinal cord without sign of microgliosis or astrogliosis.

*Therapies targeting DNA and RNA—RNA interference (RNAi):* RNA interference is a physiological mechanism, that permits to eukaryotic cells to control gene expression; this process involves small RNA (RNAs) which have a length lesser than 30 bases. Often the concerning RNAs is a double-strained small RNA (dsRNA). dsRNA less than 21–22 bases pairs (bp), which are recognized by enzymatic cascade of RNAi, are known as small interfering RNA (siRNA). The chemical bond between a siRNA and a mRNA can inactivate the expression of the mRNA target; this mechanism leads to silencing of the targeted genes. Micro RNA (miRNA) and short hairpin RNA (shRNA) are duplexes synthesized in nucleus, that inhibit the translation of targeted mRNA and promote its degradation. This mechanism was studied with success in vitro and in vivo trials since 1988 to silence the expression of mutated gene or viral genes integrated in genome of a targeted cell [117,118]. Therefore the use of siRNA is very promising in inherited diseases [119].

The goal of RNA interference (RNAi) in polyglutaminopathies is to inhibit the synthesis of defective polyglutamine-proteins resulting from mutated genes. RNAi decreases the levels of ataxin-7 in the mouse model of SCA7 [120]. RNAi improved clinical phenotype in a study using a knock-in (KI) mouse model of SCA1 [121].

#### 4.1.2. ADCAs Induced by Toxic RNAs

Toxic RNAs underlie the cellular degenerations in ADCAs caused by an expanded microsatellite in an intron (SCAs 10, 31, 36 and 3) and in 3′ untranslated region (SCA8) [103]. These diseases share common pathomechanisms in which expanded repeats, consisting of trinucleotides, pentanucleotids, or hexanucleotide units, bind to RNA binding proteins, leading to the loss of their functions or the formation of stress granule [103]. Both of them are assumed to be responsible for cell death. Thus, it could be a promising strategy to reduce the level of these toxic RNAs using ASO or RNAi technologies [103].

#### 4.1.3. ADCAs Caused by Point Mutations

When hereditary CAs are caused by point mutations, cerebellar degeneration is attributed to a novel toxic function or a dominant negative effect by the mutant protein [103]. Promising methods would include the gene silencing or the skipping of the mutated exon by ASO/RNAi [103].

Table 5 shows candidate disease-modifying and neuromodulatory drugs for ADCAs [122,123,124,125,126,127,128,129,130,131,132,133,134]. The former includes oligonucleotide therapeutics and the intervention of downstream pathways.

### 4.2. Recent Advances in Neurotransplantaion

#### 4.2.1. Therapeutic Rationale

Cerebellar transplantation is a promising therapy, especially in degenerative CAs [135,136,137,138]. Immature embryonic or stem cells are grafted to develop and integrate into the host’s tissue [135,136,137,138,139,140]. Experimental transplantation into the cerebellum of embryonic (fetal) cerebellar tissue, embryonic (fetal) or adult neural stem cells, embryonic stem cells, induced pluripotent stem cells (hiPSC), mesenchymal stem cells isolated from various tissues (e.g., bone marrow and adipose tissue), and carcinoma stem cells have already been described [135,136,137,138,139,140]. Cell neurotransplantation in the cerebellum could elicit therapeutic benefits through following three mechanisms; (1) rescue of degenerating neurons, (2) facilitation of compensation, and (3) reconstruction of damaged neural circuitries [136,137,138].

*Rescue of degenerating neurons:* Several experiments have shown that grafted cells prevent degeneration of the cerebellar neurons through neurotrophic or metabolic support to degenerating cells [141,142,143,144,145], rectification of the levels of specific pathological factors [146], and/or suppression of inflammatory reactions. Prevention of neural degeneration leads to maintenance of cerebellar reserve. For example, mesenchymal stem cells of newborn Lurcher mice grafted in the cerebellum, secreted brain-derived neurotrophic factor (BDNF), neurotrophin 3 (NT-3), and glial-derived neurotrophic factor (GDNF), and rescued PC from degeneration [141]. Furthermore, mesenchymal stem cells could suppress degenerative processes in Niemann-Pick model mice [142,143]. Grafted neural stem cells established gap junctions with host PCs [144] and grafted bone marrow-derived cells established cell fusion with host PCs [145], both of which led to the prolongation of survival period of PCs in model mice. Grafted neural stem cells reduced the level of tissue plasminogen activator, which is involved in the degenerative processes in Nervous mutant mice [146].

*Facilitation of compensation and restoration:* Transplantation facilitates cerebellar compensation, on the basis of synaptic plasticity. Previous studies showed that BDNF, secreted by the grafted mesenchymal stem cells, upregulate glutamate-containing synaptic vesicles at parallel fiber-PC synapses [147], and control GABAergic synaptic transmission [148]. These changes promote the reorganization of residual neural components so as to compensate and restore loss of other PCs, and potentiate cerebellar reserve [136,137,138].

*Reconstruction of damaged neural circuitries:* The optimal goal of treatment is substitution of severely damaged cerebellar circuitries and preservation of the cerebellar reserve [136,137,138]. No doubt that such substitution requires several stages of elaborate reconstruction; engraftment of adequate cell numbers, long-term survival, proper differentiation of grafted cells into the target cell phenotype(s), appropriate migration of grafted cells into the desired location of cell substitution, formation of synapses with their inputs and targets, and finally functional integration in the cerebellar circuitry [136,137,138]. Thus, to achieve reconstruction is more difficult compared with the other two goals. This is in contrast to transplantation in Parkinson’s disease (PD), where the grafted cells are expected to secrete dopamine, establish a simplified reconstruction in the nigro-striatal pathways, and facilitate compensatory functions in postsynaptic striatum neurons [149].

Triarhou and coworkers [150] reported successful transplantation of fetal cerebellar cell suspension into the cerebellum with partial restoration of the corticonuclear projections in PCD mice. On the other hand, many other studies concluded that cerebellar reconstruction after cell transplantation is problematic and difficult to achieve. For example, the granular layer of the cerebellar cortex acts as a barrier that prevents sufficient reconstruction of the cortico-nuclear projections [151], whereas proximity of the grafted cells and cerebellar nuclei is necessary for functional organization [152].

#### 4.2.2. Possible Indications for Neurotransplantation

Here we discuss the possible indications for neurotransplantation based on the above two realistic outcome, the rescue of degenerated cells and facilitation of compensation and restoration. Understandingly, any indication will depend on the host-related etiologies that can influence survival, cell differentiation, and functional integration [136,137,138]. In any decision making regarding neurotransplantation, other therapies should also be taken into considerations. Thus, in cases of metabolic CAs and Immune-mediated CAs, when various combinations of cause-cure treatments and immunotherapies have no benefits in the progression of the disease, cerebellar transplantation should be considered [136]. Degenerative CAs are other candidates for cerebellar transplantation. Thus, neurotransplantation can be a therapeutic option when the pathological conditions are progressive and uncontrollable. When neurotransplantation is considered a suitable option, it should be introduced when cerebellar reserve is still preserved. This is because the aim of transplantation is to maintain and enhance cerebellar reserve, and hence alleviate the disease progress [136].

The indications for neurotransplantation should also be examined from the point of negative effects and potential complications, which have been learned from transplantation in PD [138]. Transplantation of highly proliferative cells can elicit carcinogenesis [138]. For instance, human embryonic stem cells (hESCs) have high proliferative capacity and can grow rapidly to form tumor-like structure of neuroepithelial cells [153]. Although human induced pluripotent (hIPS) stem cells are promising, their use requires Myc and KLf4 reprogramming factors, but these are oncogene factors [154]. Furthermore, the non-self immature cells can elicit autoimmune responses [138]. For this reason, various combinations of immunosuppressants were used for more than six months after neurotransplantation in many open-label clinical trials of neurotransplantation in PD [155].

## 5. Conclusions

CAs represent a growingly recognized heterogeneous group of diseases from the phenotypic and pathogenic standpoint. The etiopathogenesis of several CAs is well established and effective treatments are available for some of them. For both ARCAs and SCAs, only symptomatic treatments are available for most of them. Recently, promising therapies have been assessed in cellular and animal models, especially molecules acting on the RNA machinery. Clinical studies are required to demonstrate efficacy and safety of these approaches, which may turn degenerative CAs as fully treatable disorders.

## Figures and Tables

**Figure 1 brainsci-10-00011-f001:**
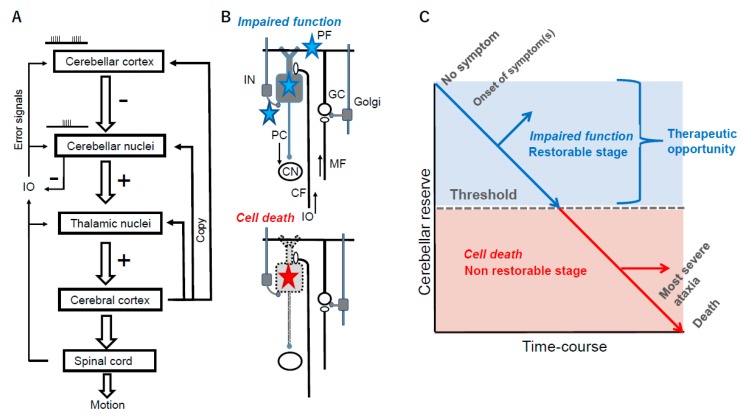
A schematic draw of cerebellar reserve. (**A**) Cerebello-cerebral loops involved in expectations and estimates of future motor/cognitive states. The cerebellar cortex inhibits (-) cerebellar nuclei via Purkinje cells (PC). Cerebellar nuclei exert and excitatory drive (+) over the thalamic nuclei. Cerebellum computes expected motor/cognitive outcomes, relayed via the cerebello-thalamo-cortical pathway. Cerebral cortex sends a copy of motor/cognitive commands to the cerebellar circuitry. The inferior olive (IO) serves as a comparator signaling errors between the expected outcome and the actual outcome via reafferent informations. The climbing fibers tune the activity of the Purkinje layer. Cerebellum re-build constantly the set of expectations in daily life. (**B**,**C**) During the initial period, cerebellar reserve is preserved. As cell death advances, cerebellar reserve is lost. PC: Purkinje cell, IN: inhibitory interneurons, Golgi: Golgi cell, GC: granule cell, PF: parallel fiber, MF: mossy fiber, CF: climbing fiber, IO: inferior olive nucleus.

**Table 1 brainsci-10-00011-t001:** Currently used therapies for metabolic and cerebellar ataxias.

Disorder	Management
**Metabolic cerebellar ataxias**	
Alcohol-related cerebellar ataxias	Abstinence and correction of malnutrition, rehabilitation
Wernicke’s encephalopathy	Replenishment of vitamin B1 using: (1) thiamine at 100 mg/day (Galvin et al., 2010 [19]), (2) thiamine at a minimum dose of 500 mg three times a day for patients with clinical features (Sechi and Serra, 2007 [15]), (3) parenteral thiamine at 200 mg for patients without apparent clinical features (Ambrose et al., 2001 [20]).
Superficial siderosis	Administration of iron chelator: deferiprone; 15 mg/kg body weight/day (Kuo et al., 2017 [25]).
**Immune-mediated cerebellar ataxias**	
Gluten ataxia	Strict gluten-free diet. If no benefits are observed, check adherence or hypersensitivity
Paraneoplastic cerebellar degeneration	Surgical excision of the tumor followed by immunotherapy: mPSL, IVIg, immunosuppressants, or/and plasma exchange
Post-infectious cerebellitis	Often self-limiting. Antibiotics in selected cases. Surgical decompression in case of herniation
Anti-GAD ataxia	Induction therapy (mPSL, IVIg, immunosuppressants, plasma exchange, or/and rituximab) followed by maintenance therapy (long-term oral PSL, IVIg, immunosuppressants, or/and rituximab)

Abbreviations: mPSL: intravenous methylprednisolone; oral PSL: oral prednisolone; IVIg: intravenous immunoglobulins; GAD: glutamate decarboxylase.

**Table 2 brainsci-10-00011-t002:** Therapies for autosomal recessive cerebellar ataxias.

Proposed Mechanism	Treatment	Efficacy
**Friedreich’s ataxia**		
Anti-oxidant	Combination of vitamin E and coenzyme Q10	*Potential therapies* A double-blind study demonstrated CAs improvement. (Artuch et al., 2002 [52], Pineda et al., 2008 [53]) However, no control placebo group was included.
	Idebenone	*Potential therapies* Two open labelled trials showed improvements or stabilization of CAs. (Artuch et al., 2002 [52], Pineda et al., 2008 [53]) However, these results were not reproduced in double-blinded placebo-controlled studies.
Chelation of accumulated iron	Deferiprone	*No evidence* Multicenter randomized placebo-control study showed deterioration of CAs with 40 and 60 mg/kg/day, and inconclusive results with 20 mg/kg/day. (Pandolfo et al., 2013 [54])
Increase in frataxin protein expression level	Interferon	*Potential therapies* Open-label trial showed subcutaneous injection of interferonγ over 12 weeks improved Friedreich’s ataxia score. (Seyer et al., 2015 [55])
**Ataxia-telangiectasia**		
Anti-oxidants	Betamethasone	*Potential therapies* A randomized control trial showed 13-point reduction in International Cooperative Ataxia Rating Scale. However, the study included only 13 patients with short-term observation (31 days). (Zannolli et al., 2012 [56])
**Ataxia with vitamin E deficiency**		
Replacement of vitamin E	Vitamin E	*Approved and supportive therapy*
**Abetalipoproteinemia**		
Low fat diet, decreased long-chain fatty acids and oral essential fatty acids Replacement of fat-soluble vitamin	Vitamin A, E, D, K	*Approved and supportive therapy*
**Wilson’s disease**		
Chelation of accumulated copper	D-penicillamine Trientine Zinc acetate/sulfate	*Approved and supportive therapy*
**GLUT1 deficiency**		
Ketogenic diet	-	*Approved and supportive therapy*
**Refsum’s diseases**		
Phytanic acid-free food	Dietary restriction	*Approved and supportive therapy*
**Cerebrotendinous xantomatosis**		
Replacement of decreased bile acid	chenodeoxycholic acid, ursodeoxycholic acid, cholic acid, and taurocholic acid	*Approved and supportive therapy*
**Niemann-Pick disease type C**		
Inhibition of glucosylceramide synthesis	Miglustat	*Approved and supportive therapy*

**Table 3 brainsci-10-00011-t003:** Protocols and outcomes of rehabilitation.

Studies	Protocols	Outcomes
**Motor rehabilitation**		
Ilg et al. (2010) [74]	16 ataxic patients (age: 61 ± 11 years, 10 patients with degenerative CAs and 6 patients with sensory ataxia, disease duration: 12.9 ± 7.8 years, baseline SARA score: 15.8 ± 4.3). 1 h × 3/week × 4 weeks Post training; home training	SARA and gait analysis improved only in patients with cerebellar ataxia not afferent ataxia After 1 year, improvements in motor performance and achievements in activities of daily life persisted
Miyai et al. (2012) [76]	42 ataxic patients (age: 62.5 ± 8.0 years, all had degenerative CAs, disease duration: 11.3 ± 3.8 years, baseline SARA score: 11.3 ± 3.8) 2 h × 5 + 1 h × 2/week × 4 weeks Post training; none	SARA and gait analysis improved. Improvement was prominent in stability than in limb coordination Gains were maintained within 6 months
**Cognitive rehabilitation**		
Maeshima and Osawa (2007) [79]	61-year-old man Disoriented in time, had problems with recent memory, attention deficits, executive dysfunctions, and poor volition and spontaneity Occupational therapy of real orientation therapy and attention process training	No improvement in executive functions or visuo-spatial orientation
Schweizer et al. (2008) [80]	41-year-old man Executive dysfunctions Goal Management Training; to resume executive and attentional control by consciously interrupting automatic behaviors	Therapeutic gain insignificant. However, the patient was able to resume professional activities due to increased awareness of shortcomings and error-prone situations.
Komuro et al. (2014) [81]	34-year-old man Impaired visuospatial cognition, attention, working memory, sensory processing, and executive function Writing, calculating, computing, and planning exercises	Improvement in the listed cognitive functions
Ruffieux et al. (2017) [82]	16-year-old man Severe motor, cognitive, and emotional disorders Emulation board; patient encouraged to cooperate with staff in a football game to receive a reward	After 2 months, improvements in executive function, attention, memory, mental processing speed, and mental arithmetic

**Table 4 brainsci-10-00011-t004:** Protocols and outcomes of noninvasive cerebellar stimulation.

Studies	Protocols	Outcomes
Shimizu et al. (1999) [89]	*n* = 4 (mean age 49 ± 24) Degenerative CAs One session; 10 stimuli of 0.1 ms each for 21 days	Pre/post Improvement in 10 m walking and increase in cerebellar blood flow
Shiga et al. (2002) [90] rTMS	*n* = 74 (mean age 58 ± 2), Degenerative CA One session; 10 stimuli of 0.1 ms each for 21 days Active/Sham-controlled	Pre/post Improvement in 10 m walking and standing
Kim et al. (2014) [91] rTMS	*n* = 32 (mean age 67 ± 10), Ischemia 15 min sessions of 1 Hz × 5, for 5 days Active/Sham controlled	Pre/post Improvement in 10 m walking
Grimaldi et al. (2013) [93] anodal tDCS	*n* = 9 (mean age 51 ± 14) 1 mA, 20 min Active/Sham controlled	Pre/post No improvement in posture and reduction of stretch reflex gains. No change in mechanical counter test.
Grimaldi et al. (2014) [86] anodal tDCS	*n* =2 (mean age 46 ± 4), Degenerative CAs 1 mA, 20 min + 20 min Active/Sham controlled	Pre/post, cerebello-cerebral stimulation Improvement in SARA, dysmetria and onset latency in antagonistic muscles (from 108–98 to 63–57 ms in patient 1, and from 74–87 to 41–46 ms in patient 2) Improvement in tremor
Benussi et al. (2015) [87] anodal tDCS	*n* = 19 (mean age 54 ± 18), Degenerative CAs 2 mA, 20 min Active/Sham controlled	Pre/post Improvement in SARA by about 10%, ICARS by 12%, ine-Hole Peg Test by 11%, 8-m walking time by 11%.
Benussi et al. (2017) [88] anodal tDCS	*n* = 20 (mean age Sham tDCS 50 ± 17; mean age Anodal tDCS 55 ± 18.2), Degenerative CAs 2 mA, 20 min Active/Sham controlled	Pre/post/Long-term follow up (4–12 weeks) Improvement in SARA by about 3%, ICARS by 12%

**Table 5 brainsci-10-00011-t005:** Candidate drugs and for autoimmune dominant cerebellar ataxias.

SCA Type	Candidate Drug	Assumed Therapeutic Rationale
**Disease-modifying drugs**		
*Oligonucleotide therapeutics*		
SCA1, 2, 3	ASO against ATXN1, ATXN2 or ATXN3	In these SCAs, toxic gain-of-function mechanisms are well established. ATX2, 3-targeting ASO ameliorated the symptoms in mouse models [113,114,115,116,122].
SCA1	AAV-mediated delivery of short hairpin RNA	RNAi improved ataxia, restored cell morphology, and decreased ataxin-1 inclusions in Purkinje cells in an SCA1 mouse model [121,123].
SCA3	Lentiviral-mediated delivery of short hairpin RNA	RNAi downregulated ATXN3 to reduce neuropathology in a SCA3 rat model [124].
SCA3	AAV-mediated delivery of micro RNA	RNAi suppresses ATXN3 levels and cleared abnormal nuclear accumulation of mutant ATAXN 3 in a SCA3 mouse model [125].
SCA6	AAV-mediated delivery of micro RNA (miR-3139-5q)	RNAi attenuates IRES-driven translation of toxic α1ACT66, protected ataxia in a SCA6 mouse model [126].
SCA7	AAV-mediated delivery of micro RNA to retina	RNAi preserved normal retinal function in a SCA7 mouse model [127].
SCA7	AAV-mediated delivery of micro RNA	RNAi suppressed ATXN7 level and improved ataxia in a SCA7 mouse model [120].
*Intervention on downstream pathways*		
SCA1	MSK inhibitor	Inhibitors of the RAS–MAPK–MSK1 pathway decreased ATXN1 levels and suppressive neurodegeneration in animal models of SCA1 [128].
SCA2	Dantrolene	Dantrolene inhibited intracellular Ca2^+^ release and protected Purkinje cells from cell death in an SCA2 mouse mode [129].
SCA3	Dantrolene	Dantrolene inhibited intracellular Ca2^+^ release and protected neuronal cells in pontine nuclei and substantia nigra regions from cell death in SCA3-YAC-84Q transgenic mice [130].
SCA3	Citalopram	Citalopram, a selective serotonin reuptake inhibitor, inhibited mutant ATXN3 aggregation and reduced ATXN3 neurotoxicity through neuronal serotonin pathways in cells and a SCA3 mouse mode [131].
SCA3	Aripiprazole	Aripiprazole reduced mutant ATXN3 levels in a cell-based assay [132].
**Neuromodulation therapies**		
SCA6 and other SCAs	4-Aminopyridine	4-Aminopyridine, a nonselective blocker of the Kv family of K channels, restore pacemaker activities of Purkinje cells. Efficacies in ocular disorders were reported [4].
SCA2 and other SCAs	Chlorzoxazone	Chlorzoxazone, a small-conductance calcium-activated potassium channel activator, normalizes of the Purkinje cell spontaneous activities [133].
SCA44	Nitazoxanide	Nitazoxanide, a negative allosteric modulator of metabotropic glutamate receptor 1 and 5, inhibited mutant forms of these receptors in transfected cells [134].

AAV: adeno-associated virus, ASO: antisense oligonucleotide, RNAi: RNA interference, IRES: internal ribosomal entry site, MSK: mitogen- and stress-activated protein kinase.

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
