# Peer review of "Recent Advances in the Treatment of Cerebellar Disorders"

_brainsci, 2019, doi:10.3390/brainsci10010011_

Round 1

Reviewer 1 Report

The manuscript by Mitoma et al. reviews current treatments used for a wide range of cerebellar disorders.

The review is both informative and very well written. A plethora of cerebellar disorders and current/experimental treatments are covered within the manuscript.

Upon reflection, my single suggestion is that it would be helpful to add a section on the causes and potential treatments of the Spinocerebellar ataxias (SCAs) – particularly as several SCAs are defined as ‘pure’ cerebellar ataxias. 

Author Response

We are grateful to the reviewer 1 for the evaluation of the manuscript.

We agree with the comment. We added a new Table sumamrizing SCD therapies  available or developement (Table 5). Accordingly, we added new references and made some modifications in Text (highlighted by yellow marker).

Reviewer 2 Report

The authors have submitted an interesting review regarding most of the available and potential treatments for different types of cerebellar disorders.

The authors have submitted an interesting review regarding most of the available and potential treatments for different types of cerebellar disorders.

The manuscript is well written, clearly organized, and the bibliography is quite actual end extensive. There is not any suggestion to improve the quality of this manuscript

Author Response

We are grateful to the reveiwer 2 for the evaluation of the manuscript.